# Serum Uromodulin, a Potential Biomarker of Tubulointerstitial Damage, Correlates Well with Measured GFR and ERPF in Patients with Obstructive Nephropathy

**DOI:** 10.3390/medicina58121729

**Published:** 2022-11-26

**Authors:** Marija Vukmirović Papuga, Zoran Bukumirić, Branislava Ilinčić, Romana Mijović, Tanja Šašić Ostojić, Radmila Žeravica

**Affiliations:** 1Department of Pathophysiology and Laboratory Medicine, Faculty of Medicine, University of Novi Sad, 21000 Novi Sad, Serbia; 2Center of Laboratory Medicine, Clinical Center of Vojvodina, 21000 Novi Sad, Serbia; 3Institute of Medical Statistics and Informatics, Faculty of Medicine, University of Belgrade, 11000 Belgrade, Serbia

**Keywords:** chronic kidney disease, effective renal plasma flow, glomerular filtration rate, kidney function, obstructive nephropathy, serum uromodulin

## Abstract

*Background and Objectives*: In chronic kidney obstruction, the severity of tubulointerstitial damage correlates best with the loss of kidney function and the risk for progression to end-stage kidney disease. The present study aimed to investigate the potential clinical significance of serum uromodulin (sUmod) as a marker of early kidney disfunction in patient with obstructive nephropathy (ON). *Materials and Methods*: Serum Umod level was measured by sensitive ELISA method in 57 adult patients with obstructive nephropathy and 25 healthy subjects in control group. Kidney function was precisely evaluated via measured glomerular filtration rate (mGFR) (renal clearance of 99 mTc-diethylenetriamine penta-acetic acid), effective renal plasma flow (ERPF) and Cystatin C level. Recruited patients were divided into subgroups based on the mGFR: group I—GFR ≤ 60 mL/min/1.73 m^2^ (N = 31), group II—GFR > 60 mL/min/1.73 m^2^ (N = 26). *Results*: A significantly lower level of serum uromodulin was measured in patients with ON (50.2 ± 26.3 ng/mL) compared to the control group (78.3 ± 24.5 ng/mL) (*p* < 0.001). The mean level of serum Umod was significantly different between group I (30.5 ng/mL ± 11.1) and group II (73.6 ng/mL ± 18.6) (*p* < 0.001), but not between group II (73.6 ng/mL ± 18.6) and control group (78.3 ± 24.5 ng/mL). There was a positive correlation between sUmod and mGFR (R = 0.757, *p* < 0.001) and ERPF (R = 0.572 *p* < 0.001), with lower sUmod levels in patients with impaired renal function. An inverse relationship was detected between sUmod and filtration markers—cystatin C (R = −0.625, *p* < 0.001), creatinine, urea and uric acid. ROC analysis of sUmod to differentiate between ON patients with GFR below 60 mL/min/1.73 m^2^ and above 60 mL/min/1.73 m^2^ resulted in AUC of 0.98 (*p* < 0.001, 95% CI 0.922 vs. 0.998) at a cut-off value of 46 ng/mL (specificity 96.8%, sensitivity 92.2%). *Conclusions*: The significant correlation of sUmod with kidney function parameters may imply potential clinical significance as a noninvasive biomarker of early kidney disfunction in obstructive nephropathy.

## 1. Introduction

Urinary tract obstruction (UTO) is a common urological condition caused by a variety of diseases that may occur in patients of all ages. Obstructive uropathy that leads to obstructive nephropathy (ON) is, although not the most frequent, an important cause of chronic kidney disease (CKD) [1]. It is known that chronic obstruction induces tubular and interstitial injury by intrarenal angiotensin II activation followed by the release of cytokines and adhesion molecules, in turn, triggering macrophage infiltration, the production of reactive oxygen species and decreased renal blood flow [2,3]. Independent of the primary lesion, the severity of tubulointerstitial damage correlates best with the loss of kidney function and the risk for progression to end-stage kidney disease.

Prolonged obstruction of the urinary tract may cause atrophy of the renal parenchyma. Due to the lack of symptoms in the early stages of the chronic obstruction, renal function is already partially reduced when symptoms appear [4].

Biomarkers of kidney function might help in detection of early stages of kidney failure. The most commonly used biomarker to evaluate global kidney function is the serum creatinine level. However, it is known that the serum creatinine level is insensitive to early changes in the GFR and unilateral kidney damage [5]. NGAL, KIM-1 and NAG were studied as biomarkers in ONs, such as ureteropelvic junction obstruction (UPJO) and ureteral calculi, and are regarded as promising early biomarkers of progressive renal damage. However, the clinical usefulness of these biomarkers is limited by insufficient sensitivity or specificity [5]. Early diagnosis of ON might improve patient outcomes by allowing timely and targeted therapy to be implemented, as well as reducing the incidence of adverse outcomes such as progression to end stage renal disease, diabetes and cardiovascular risk [1]. Currently used imaging techniques and conventional laboratory parameters are insufficient to assess the early onset of this condition. Therefore, there is a constant demand for new biomarkers helpful in stratification and monitoring of patients with ON.

One of the possible biomarker candidates might be uromodulin (Umod), a protein mainly produced by the thick ascending limb of the loop of Henle and, to a lesser extent, in the initial segment of the distal convoluted tubule [6,7].

Despite its discovery more than 60 years ago, the primary physiological function of Umod is still not completely understood. Available data suggest that this protein might be of importance in the regulation of salt transport, protection against urinary tract infection and kidney stone formation, and might also participate in kidney injury and innate immunity [8]. The important role of Umod in patients both with acute and chronic kidney diseases was demonstrated in previous studies [9,10]. Apart from being present in urine, it can also be found in serum, and both urine and serum concentrations are directly associated with kidney function [11]. Based on evidence that tubular function is impaired early in the disease course, even before evident glomerular dysfunction, Umod may represent a promising biomarker of tubular integrity.

The present study aimed to investigate the potential clinical significance of serum uromodulin (sUmod) as a marker of early kidney disfunction in patient with CKD caused by ON. Therefore, we correlated the sUmod level with kidney functional parameters, a representative and highly accurate measurement of the glomerular (measured GFR) and tubular function (measured ERPF), as well as with Cystatin C, the most sensitive marker of kidney damage.

## 2. Materials and Methods

The cross-sectional study conducted at the Department of Nuclear Medicine, Center of Laboratory Medicine, Clinical Center of Vojvodina (CCV), from 2019 to 2022, was approved by the Ethics Committee of CCV in adherence with the Declaration of Helsinki. Written informed consent was obtained from all study participants.

### 2.1. Study Subjects

The study population included 57 patients diagnosed with CKD caused by ON. They were referred to the Department of Nuclear Medicine for renal scan and radioisotope clearance to evaluate total and split renal function. All the included patients were diagnosed to have CKD by means of clinical examination, biochemical analysis, and imaging abnormalities according to the K/DOQI criteria [12]. According to the documentation supplied with the referrals, the etiology of ON was as follows: urolithiasis (41% of the patients); intrinsic ureteric stricture (29% of the patients); ureteropelvic junction obstruction (20% of the patients); and unknown etiology (10% of the patients).

Recruited patients were further divided into two groups based on the measured GFR values: GFR ≤ 60 mL/min/1.73 m^2^ (N = 31) or GFR > 60 mL/min/1.73 m^2^ (N = 26). Exclusion criteria for patients were: measured GFR (mGFR) < 15 mL/min/1.73 m^2^, patient with solitary kidney, diabetes mellitus, chronic liver diseases, acute inflammatory and infectious diseases and malignant diseases.

The control group consisted of 25 healthy individuals, potential kidney donors. Their renal function was evaluated by renal scan and clearance in accordance with the set algorithm for examination of renal function. They were referred to the Department of Nuclear Medicine by Department of Nephrology and Clinical Immunology of Clinical Center of Vojvodina.

### 2.2. Study Protocol

All subjects were physically examined. Body height was measured by Harpenden anthropometer (Holtain Ltd., Croswell, UK) to the nearest 0.1 cm, while body weight was measured in the standing upright position with a Seca scale (model 700, Hamburg, Germany). BMI was calculated as the ratio of body weight to the square of body height (kg/m^2^). Blood and urine samples were taken on the same day. The GFR and ERPF were measured in all study subjects.

### 2.3. Measurement of Uromodulin Concentration

Fasting venous blood samples were obtained at the study visit. All participants’ blood samples were stored at −80 °C until use. Serum Umod was measured by a commercially available human uromodulin ELISA kit (CLOUD-CLONE CORP.) according to the manufacturer’s instructions.

### 2.4. Assessment of Renal Function

In order to accurately determine renal function, the following parameters were measured:

(a) GFR measurement

GFR was determined (using the single-compartment model) via isotopic clearance of 99 mTc-labeled diethylenetriamine penta-acetic acid (99 mTc-DTPA) following a single injection of 37 MBq. Two venous blood samples were drawn 180- and 240-min post injection [13]. The GFR values were expressed as ml/min/1.73 m^2^.

(b) ERPF measurement

ERPF was determined via isotopic clearance of 131I-labeled orthoiodohippuric acid (hippuran) by Blaufox method, in two blood samples collected 20 and 30 min after injection [14]. The ERPF values were expressed as ml/min/1.73 m^2^.

Quality control of the radiochemical purity of isotopes (>95%) was performed with paper chromatography. A well gamma counter with NaI (Tl) crystal (Captus 3000 by Capintec, USA) was used to measure the radioactivity of samples.

(c) Biomarkers of renal function

The measurements of cystatin C concentration were performed by the immunoturbidimetric method on Mindray BS 2000 (commercial set by Mindray China, reference range: 0.62 to 1.15 mg/l). Serum and urine levels of creatinine, as well as and serum urea and uric acid concentrations were determined by standard biochemical methods using commercial kits on analyzer Advia 1800 (Siemens, Germany). Urinary albumin concentration was measured by nephelometric immunoassay.

In order to exclude patients with diabetes, the fasting blood glucose level was measured using the glucose hexokinase method, as well as blood insulin level using an electrochemiluminescence immunoassay on ADVIA Centaur XP immunoassay system (Siemens, Germany).

### 2.5. Statistical Analysis

Depending on the type of variables and the normality of the distribution, the data description was displayed as n (%), mean ± standard deviation or median (range). The correlation between variables was estimated using Pearson’s and Spearman’s correlation coefficient. A linear regression model was used for modelling the relationship of dependent variables with potential predictors. The predictors from univariate analyses that were statistically significant at a significance level of 0.05 were included in the multivariate regression models. All *p* values less than 0.05 were considered significant. Uromodulin diagnostic performances in predicting GFR of less than 60 mL/min/1.73 m^2^ and GFR of less than 90 mL/min/1.73 m^2^ with optimal level of specificity and sensitivity were evaluated in receiver-operating curve (ROC)-analysis in order to determine the cut-off point. All statistical analyses were performed using the IBM SPSS Statistics 22 (SPSS Inc., Chicago, IL, USA) software package.

## 3. Results

### 3.1. Baseline Characteristics

General characteristics of patients with ON and control subjects are provided in Table 1. The median age in patients with ON was 59.6 ± 13.1 years and 53.8 ± 11.7 years in control group—participants with GFR > 90 mL/min/1.73 m^2^. There were no significant differences in age, gender and BMI between the groups. Measured GFR and ERBF in control group were significantly higher compared to participants with ON.

### 3.2. Renal Function Parameters

Recruited patients with ON were further divided into two groups based on the measured GFR values: group I GFR ≤ 60 mL/min/1.73 m^2^ (N = 31); group II GFR > 60 mL/min/1.73 m^2^ (N = 26). Renal function parameters of patients with ON and participants of the control group are given in Table 2.

There was significant difference detected in mGFR and ERPF values among ON patients with different degrees of renal hypofunction and healthy subjects (*p* < 0.001). Significantly higher serum concentrations of creatinine (group I 132.8 ± 37.9 µmol/L; group II 73.9 ± 14.5 µmol/L; control group 71.9 ± 14.8 µmol/L), urea (group I 8.6 ± 3.4 mmol/L; group II 4.9 ± 1.6 mmol/L; control group 1.5 ± 1.2 mmol/L) and uric acid (group I 396.1 ± 100.8 µmol/L; group II 345.4 ± 85.5 µmol/L; control group 299.2 ± 82.4 µmol/L) were detected in group I compared to group II and control group (*p* < 0.001). Although lower serum concentrations in all three parameters were measured in control group compared to group II, difference was not statistically significant. A similar pattern was observed in serum cystatin C concentration with significant difference between group I (1.7 ± 0.5 mg/L) and group II (1.0 ± 0.2 mg/L) (*p* < 0.001), and an insignificant difference between group II and control group (0.8 ± 0.1 mg/L).

### 3.3. Serum Uromodulin

A significantly lower level of serum uromodulin was measured in patients with ON (50.2 ± 26.3 ng/mL) compared with the control group (78.3 ± 24.5 ng/mL) (*p* < 0.001). The difference in mean level of sUmod was also significantly different between group I (30.5 ng/mL ± 11.1) and group II (73.6 ng/mL ±18.6) (*p* < 0.001), but not between group II (73.6 ng/mL ± 18.6), and control group (78.3 ± 24.5 ng/mL) (Table 2).

The correlation parameters of sUmod and renal function parameters are given in Table 3. Serum Umod was significantly correlated with mGFR, ERPF, cystatin C, creatinine and urea and uric acid (Table 3). There was positive correlation between sUmod and mGFR and ERPF (R = 0.757, *p* < 0.001 and R = 0.572, *p* < 0.001), with lower sUmod levels in patients with impaired renal function (Figure 1). On the other hand, an inverse relationship was detected between sUmod and filtration markers—cystatin C (R = −0.625, *p* < 0.001), creatinine, urea and uric acid. While sUmod concentration analogues to decrease in renal function, it behaves in an opposite manner in relation to the renal filtration markers.

A significant association between sUmod concentration and mGFR was also detected in the linear regression model (*p* < 0.001) (Figure 1).

In a univariate analysis, a significant association was detected between sUmod level and age (*p* = 0.004), gender (*p* = 0.036), total cholesterol (*p* = 0.038) and fasting glucose level (*p* = 0.001) (Table 4). However, in a multivariate analysis, only fasting glucose level was found to be a significant predictor of sUmod concentration. Higher concentrations of fasting glucose in the studied population are associated with lower concentrations of sUmod.

An ROC analysis of sUmod to differentiate between ON patients with GFR below 60 mL/min/1.73 m^2^ and above 60 mL/min/1.73 m^2^ resulted in an AUC of 0.98 (*p* < 0.001, 95% CI 0.922 vs. 0.998) at a cut-off value of 46 ng/mL with a diagnostic specificity of 96.8% and sensitivity of 92.2% (Figure 2a). When the ROC analysis of sUmod was performed to differentiate between group II (ON patients with GFR 60–90 mL/min/1.73 m^2^) and control group, it resulted in an AUC of 0.78 (*p* < 0.001, 95% CI 0.676 vs. 0.865) at a cut-off value of 50 ng/mL with a diagnostic specificity of 57.9% and sensitivity of 92.0% (Figure 2b).

## 4. Discussion

The well timed diagnosis of renal damage is of highest importance for early treatment of patients with ON. Early diagnosis enables significant slowdown or complete stop of the chronic kidney disease progression to end-stage renal failure [15].

Urinary as well as serum proteins provide information on renal function and have the potential to be used as prognostic tools for early disease detection and the choice of the optimal treatment and monitoring. In recent years, Umod has emerged as a novel biomarker for earlier detection of kidney disease [16]. In a study by Garimella et al., low urinary uromodulin (uUmod) concentrations in spot urine were associated with risk of progressive kidney disease, and uUmod was found to be superior compared to established markers of renal disease [17]. On the other hand, sUmod concentrations also tend to decrease in accordance with estimated GFR and with biopsy-classified stages of renal failure in patients with CKD stages 1–5, patients with autoimmune kidney diseases and recipients of a renal allograft [18]. Lower serum and plasma uromodulin levels are also found to be associated with decreased kidney function in subjectively healthy elderly (>60 years) patients without diabetes [19].

Our study focused on the serum level of Umod in patients with ON. In ON, the pathogenic effect of the increase in intratubular hydrostatic pressure is mediated by three mechanisms: hypoperfusion induced tubular ischemia, mechanical stretching or compression of tubular cells caused by pressure and altered urinary shear stress [20]. Since, Umod is a major protein synthetized and secreted by the renal medulla, and since medullary hypoxic injury is crucial in kidney diseases, it was expected that serum uromodulin decreases in patients with chronic tubulointerstitial damage such as ON.

In the current study, the sUmod level was measured in patients with chronic ON with different degrees of GFR reduction, and the association of sUmod with renal function parameters was determined. Accurate measurements were used to evaluate glomerular and tubular function (by radioisotope clearance method), as well as the sensitive routine clinical markers of renal damage such as cystatin C. Results of our study demonstrated that patients with CKD caused by ON are characterized by lower levels of sUmod compared to healthy subjects. In addition, significant differences in sUmod level were detected between the groups with different renal function status.

To our knowledge, there are not many published studies examining the relation between Umod and renal function in patients with ON. Until now, most of the studies on tubular or early renal damage focused on urinary biomarkers [21,22,23]. Most of the authors suggest that it is insufficient to test a single urinary biomarker. In infants with congenital UTO, the role of several renal biomarkers (urinary NGAL, urinary KIM-1, urinary cystatin C, serum creatinine and serum cystatin C) to predict the need of surgery was evaluated, and the combinations of these biomarkers were superior compared to single biomarkers, with AUC-ROCs being highest for urinary NGAL + urinary cystatin C + serum cystatin C [24]. In another study, in children with congenital ureteropelvic junction obstruction, assessment of both urinary NGAL and MCP-1 was superior in diagnostic performance when compared to the separate assessment of each of two biomarkers [25].

The necessity for the combined assessment of several urinary biomarkers in the early detection of kidney damage in the upper UTO is explained by multifactorial etiology of obstruction making it impossible to diagnose all obstructions using just one biomarker [26]. In a review paper by Washino et al., the authors concluded that the problem with diagnostic potential of urinary biomarkers in ON is caused by the unilateral nature of obstructions of the upper urinary tract in most patients. Unilateral reduction in glomerular filtration combined with obstruction decreases the amount of any biomarker in bladder urine, resulting in lower biomarker AUC-ROCs in bladder urine when compared to renal pelvic urine. The authors suggest combined evaluation of both serum and bladder biomarker levels in patients with upper UTO, also mentioning that serum levels of biomarkers are less studied compared to urine values [5].

The diagnostic accuracy of sUmod tends to be superior to uUmod. In a study by Steubl et al., uUmod level correlation with estimated GFR (r = 0.581, *p* < 0.001) was inferior to plasma uromodulin (r = 0.786, *p* < 0.001). In addition, sUmod was found to be superior to discriminate between non-CKD patients and patients with first stage CKD compared to uUmod, which was not able to differentiate between non-CKD and first stage CKD subjects [27].

Considering the presumed superiority, we focused on sUmod and its diagnostic and prognostic potential in patients with obstructive uropathy. In our cross-sectional study, a significant correlation of sUmod level and kidney function parameters, foremost measured GFR and ERPF but also cystatin C, urea, creatinine, and uric acid was found. In patients suffering from ON with a higher degree of GFR and ERPF reduction, lower Umod concentrations were measured.

In a univariate analysis, a significant association was detected between sUmod level and age, gender, cholesterol level and fasting glucose level, with only fasting glucose level being a significant predictor of sUmod concentration in multivariate analysis. Higher concentrations of fasting glucose in non-diabetic patients with ON were associated with lower concentrations of sUmod. This finding is in line with other study reports and supported with the fact that hyperglycemia is known to contribute to both the onset and progression of diabetic kidney disease [28,29].

Our study data were also analyzed to determine the cut-off values of sUmod to differentiate between the healthy population and patients with CKD. A serum Umod level lower than 50 ng/mL suggests a GFR value below 90 mL/min (specificity of 57.9%, sensitivity of 92.0%), while an sUmod level lower than 46 ng/mL suggests a GFR value below 60 mL/min (specificity 96.8%, sensitivity 92.2%). The pretty narrow range between cut-off values of sUmod might be explained by the fact that Umod production reaches plateau beyond GFR of 90 mL/min [30].

The role of uromodulin as a biomarker of kidney injury was recently reviewed, concluding that serum uromodulin levels represent a valuable early predictor of renal, primarily tubular damage and that lower concentrations have a prognostic role in predicting the progression of kidney damage [16,31]. In a study by Jotwani et al., findings confirmed the importance of impaired kidney tubule function as a risk factor for subsequent kidney disease progression. According to results of the study, a 2-fold higher uUmod level was associated with the slower annual decline of estimated GFR [32]. Our study results add new information to the pool of data on sUmod levels in tubular damage and CKD. However, our study is not without limitations. Foremost, the limited sample size should be considered. Apart from sample size, the absence of pathohistological findings in assessment of the association of sUmod with the degree of histological damage should be mentioned as well. Although a higher level of uUmod was recently found to be independently associated with lower tubulointerstitial fibrosis in both human kidney biopsies and a mouse model of fibrosis [33], pathohistological findings would add to the quality of our study. On the other hand, the strength of our study is the accurate measurement of global kidney function (mGFR and ERPF), thus, avoiding disadvantages of estimated kidney function. In addition to measured kidney function, we determined the sUmod level with a sensitive and robust ELISA method, thus, avoiding pre-analytical disadvantages of urinary uromodulin.

## 5. Conclusions

The results we obtained demonstrated a significant correlation of serum uromodulin with kidney functional parameters and may imply potential clinical significance as a noninvasive biomarker of early kidney disfunction in CKD caused by obstructive nephropathy. Since the dynamic nature of Umod expression and secretion is well known, as well as it is known to change rapidly in response to different pathophysiological conditions, with serum Umod levels being less well established, further investigation is necessary. The primary focus of further research should be the evaluation of the value of sUmod in accurate selection of the patients who would benefit from timely intervention and, thus, prevent significant loss of renal function.

## Figures and Tables

**Figure 1 medicina-58-01729-f001:**
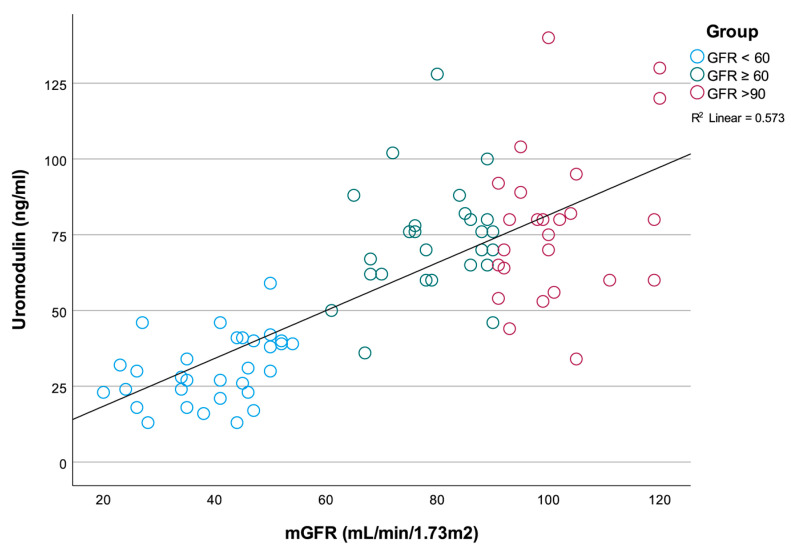
Correlation between sUmod concentration and mGFR.

**Figure 2 medicina-58-01729-f002:**
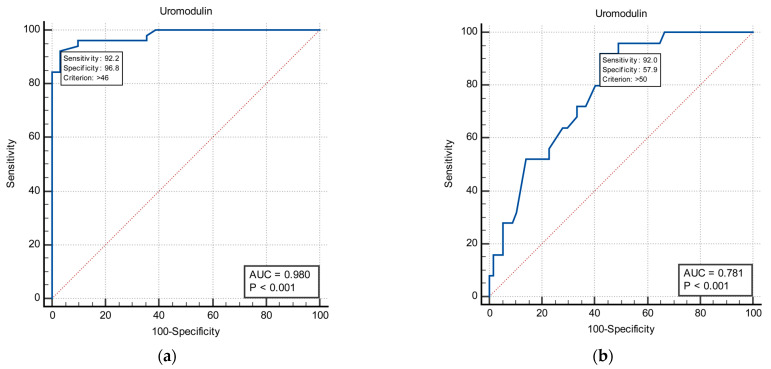
ROC analyses for the predictive value of sUmod for mGFR < 60 mL/min/1.73 m^2^ (**a**), and mGFR < 90 mL/min/1.73 m^2^ (**b**).

**Table 1 medicina-58-01729-t001:** Baseline characteristics of study population.

	ON Patients N = 57	Control Subjects N = 25	*p* Value
Number F/M, N	32/25	18/7	0.175
Age (years), mean ± SD	59.6 ± 13.1	53.8 ± 11.7	0.064
BMI (kg/m^2^), mean ± SD	28.7 ± 5.0	26.9 ± 7.6	0.288
W/H ratio (cm), mean ± SD	0.95 ± 0.13	0.91 ± 0.01	0.720
Fasting glucose level (mmol/L), mean ± SD	5.4 ± 1.2	5.0 ± 0.6	0.219
Insulin (mlU/L), median (range)	14.3 (3.0–50.2)	10.9 (8.0–30.5)	0.771
Cholesterol (mmol/L), mean ± SD	5.2 ± 1.2	5.5 ± 0.7	0.635
HDL (mmol/L), mean ± SD	1.3 ± 0.4	1.3 ± 0.5	0.983
LDL (mmol/L), mean ± SD	3.1 ± 0.8	2.7 ± 1.1	0.466
Triglycerides (mmol/L), median (range)	1.6 (0.7–3.6)	2.5 (0.7–5.2)	0.440
C-reactive protein (mg/dL), median (range)	2.4 (0.4–11.0)	1.9 (0.9–5.8)	0.646
mGFR (mL/min/1.73 m^2^), mean ± SD	57.8 ± 22.1	101.4 ± 9.6	<0.001
ERPF (mL/min/1.73 m^2^)	305.8 ± 113.4	449.8 ± 59.0	<0.001
C Cystatin C (mg/L)	1.3 ± 0.5	0.8 ± 0.1	<0.001
Uromodulin (ng/mL)	50.2 ± 26.3	78.3 ± 24.5	<0.001

BMI—body mass index; W/H—waist to hip ratio; mGFR—measured glomerular filtration rate; ERPF—effective renal plasma flow.

**Table 2 medicina-58-01729-t002:** Renal function parameters of study population.

	Group I N = 31	Group II N = 26	Control Group N = 25	*p* Overall	Group I vs. Group II	Group I vs. Control Group	Group II vs. Control Group
Number (F/M), N	14/17	18/8	18/7	0.071	0.007	0.044	0.828
mGFR (mL/min/1.73 m^2^), mean ± SD	39.7 ± 9.9	79.5 ± 9.1	101.4 ± 9.6	<0.001	<0.001	<0.001	<0.001
ERPF (mL/min/1.73 m^2^), mean ± SD	251.6 ± 60.2	393.8 ± 12.6	449.8 ± 59.0	<0.001	<0.001	<0.001	0.203
S Creatinine (µmol/L), mean ± SD	132.8 ± 37.9	73.9 ± 14.5	71.9 ± 14.8	<0.001	<0.001	<0.001	0.959
Urea (mmol/L), mean ± SD	8.6 ± 3.4	4.9 ± 1.6	1.5 ± 1.2	<0.001	<0.001	<0.001	0.828
Uric Acid (µmol/L), mean ± SD	396.1 ± 100.8	345.4 ± 85.5	299.2 ± 82.4	<0.001	0.096	<0.001	0.171
Cystatin C (mg/L), mean ± SD	1.7 ± 0.5	1.0 ± 0.2	0.8 ± 0.1	<0.001	<0.001	<0.001	0.254
Uromodulin (ng/mL), mean ± SD	30.5 ± 11.1	73.6 ± 18.6	78.3 ± 24.5	<0.001	0.007	0.044	0.828

Group I (GFR ≤ 60 mL/min/1.73 m^2^); Group II (GFR > 60 mL/min/1.73 m^2^); Control group GFR > 90 mL/min/1.73 m^2^); mGFR—measured glomerular filtration rate; ERPF—effective renal plasma flow.

**Table 3 medicina-58-01729-t003:** Correlation between serum uromodulin and renal function parameters.

	r	*p*
mGFR (mL/min/1.73 m^2^)	0.757	<0.001
ERPF (mL/min/1.73 m^2^)	0.572	<0.001
Cystatin C (mg/L)	−0.625	<0.001
Urea (mmol/L)	rho = −0.601	<0.001
S Creatinine (µmol/L)	rho = −0.644	<0.001
Uric Acid (µmol/L)	rho = −0.325	0.003

mGFR—measured glomerular filtration rate; ERPF—effective renal plasma flow.

**Table 4 medicina-58-01729-t004:** Multivariate analysis with sUmod being the dependent variable.

	Univariate		Multivariate	
	B	*p*	B	*p*
Age (years)	−0.758	0.002	0.004	0.988
Sex (Female/Male)	13.705	0.036	7.907	0.148
Cholesterol (total) (mmol/L)	5.366	0.038	3.184	0.223
HDL (mmol/L)	4.277	0.560		
LDL (mmol/L)	2.560	0.474		
Triglycerides (mmol/L)	9.621	0.052		
ApoA-I (g/L)	16.983	0.062		
ApoB (g/L)	30.046	0.026		
CRP (µg/mL)	3.231	0.146		
Blood glucose level (mmol/L)	−11.671	0.001	−4.909	0.027
BMI (kg/m^2^)	−0.531	0.425		
Waist/Hip ratio	−26.693	0.342		

## Data Availability

Not applicable.

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
