# Peer review of "Serum Uromodulin, a Potential Biomarker of Tubulointerstitial Damage, Correlates Well with Measured GFR and ERPF in Patients with Obstructive Nephropathy"

_medicina, 2022, doi:10.3390/medicina58121729_

Round 1

Reviewer 1 Report

This study investigated the sUmod level with several kidney functional parameters in patient with CKD caused by obstructive nephropathy. I have some comments to the current version.

1. The current background is insufficient, and more reasons and references should be provided to explain why obstructive nephropathy was chosen, as it is a cause of CKD with relatively small proportions.

2. Please provide a description of the study population recruitment method. Where did healthy individuals in the control group come from?

3. Table 2, have the authors made multiple comparisons? This should be further done.

4. Table 3, what are the distributions of these renal function parameters? If there are non normally distributed variables, spearman correlation coefficient should be considered.

5. There is no result of multivariate regression in this article, which should be provided.

6. I suggest that the structure of the whole discussion be reorganized, because the current logic is confusing. Please avoid introducing relevant research one after another and make appropriate summary.

Author Response

Point 1: The current background is insufficient, and more reasons and references should be provided to explain why obstructive nephropathy was chosen, as it is a cause of CKD with relatively small proportions.

Response 1: Additional reasons and references to explain why obstructive nephropaty was chosen were provided in Introduction section.

In our previous clinical studies we focused on determination of the Cystatin C levels in the blood, and measured GFR in order to determine the presence of subclinical cardiovascular disease in patients with unresponsive hypertensive phenotype (Čabarkapa et al, 2017). We also performed clinical studies focusing on the determination of concentration of the endothelin-1 in blood, and it’s importance in early detection of renal dysfunction in patients with diabetic nephropathy which is leading cause of CKD (Žeravica et al, 2015). 

In our present study, we focused on determination of the levels of serum Uromodulin. Our literature search showed that serum Uromodulin might be a significant predictor of renal dysfunction as well as primary marker of tubular dysfunction. Therefore, we selected a group of patients diagnosed with obstructive nephropathy, which are in significant number being referred to our department for evaluation renal function. Since the quest for sufficiently sensitive and specific biomarkers of renal dysfunction of obstructive etiology is still ongoing, we decided to focus on serum Uromodulin in obstructive nephropathy patients.

  1. Čabarkapa V, Ilinčić B, Đerić M, Vučaj Ćirilović V, Kresoja M, Žeravica R, Sakač V. Cystatin C, vascular biomarkers and measured glomerular filtration rate in patients with unresponsive hypertensive phenotype: a pilot study. Ren Fail. 2017 Nov;39(1):203-210. doi: 10.1080/0886022X.2016.1256316.
  2. Žeravica R, Čabarkapa V, Ilinčić B, Sakač V, Mijović R, Nikolić S, Stošić Z. Plasma endothelin-1 level, measured glomerular filtration rate and effective renal plasma flow in diabetic nephropathy. Ren Fail. 2015 May;37(4):681-6. doi: 10.3109/0886022X.2015.1010990.

Point 2: Please provide a description of the study population recruitment method. Where did healthy individuals in the control group come from?

Response 2:

Manuscript version 1 - lines 95-99.

The study population included 57 patients with CKD caused by obstructive nephropathy (ON) that were sent to Department of Nuclear Medicine for renal scan and radioisotope clearance to evaluate total and split renal function. The control group included 25 healthy individuals, whose renal function was evaluated for possible kidney donation.

were replaced with:

The study population included 57 patients diagnosed with CKD caused by ON. They were referred to the Department of Nuclear Medicine for renal scan and radioisotope clearance to evaluate total and split renal function. All included patients were diagnosed to have CKD by means of clinical examination, biochemical analysis, and imaging abnormalities according to the K/DOQI criteria [12]. According to the documentation supplied with the referrals, the etiology of ON was as follows: urolithiasis (41% of the patients); intrinsic ureteric stricture (29% of the patients); ureteropelvic junction obstruction (20% of the patients); and unknown etiology (10% of the patients).

The control group consisted of 25 healthy individuals, potential kidney donors. Their renal function was evaluated by renal scan and clearance in accordance with the set algorithm for examination of renal function. They were referred to Department of Nuclear Medicine by Department of Nephrology and Clinical Immunology of Clinical Center of Vojvodina.

Point 3. Table 2, have the authors made multiple comparisons? This should be further done.

Response 3: Multiple comparisons were made and results are added to Table 2.

Point 4. Table 3, what are the distributions of these renal function parameters? If there are non normally distributed variables, spearman correlation coefficient should be considered.

Response 4: Data/results presented in Table 3 were subjected to repeated statistical analysis. It was found that Urea (mmol/L); S Creatinine (µmol/L) and Uric Acid (µmol/L) values did not follow normal distribution. In these 3 variables spearman (instead of pearson), correlation coefficient was calculated.

Manuscript version 1 - lines 150-152

Among the methods for correlation analysis, the Pearson linear correlation coefficient was applied.

Were replaced with:

The correlation between variables was estimated using Pearson’s and Spearman’s correlation coefficient.

Point 5. There is no result of multivariate regression in this article, which should be provided.

Response 5: Results of multivariate regression analysis were added and presented in new table (Table 4).

Point 6. I suggest that the structure of the whole discussion be reorganized, because the current logic is confusing. Please avoid introducing relevant research one after another and make appropriate summary.

Response 6: The discussion section was reorganized in more logical manner. Relevant research was summed to avoid introducing research one after another. 

Point 7. English language and style are fine/minor spell check required

Response 7: The text was spell checked, as suggested.

Reviewer 2 Report

I believe that the proposed study requires a greater definition of the times of onset and uniformity of the types of obstructive nephropathy of which patients are affected. in any case, the role of uromodulin has already been examined and correlated with histopathological data. Perhaps bibliographic entries should be mentioned (e.g. Urine Uromodulin as a Biomarker of Kidney Tubulointerstitial Fibrosis)

Author Response

Point 1: I believe that the proposed study requires a greater definition of the times of onset and uniformity of the types of obstructive nephropathy of which patients are affected.

Response 1:

Manuscript version 1 - lines 95-99.

The study population included 57 patients with CKD caused by obstructive nephropathy (ON) that were sent to Department of Nuclear Medicine for renal scan and radioisotope clearance to evaluate total and split renal function. The control group included 25 healthy individuals, whose renal function was evaluated for possible kidney donation.

were replaced with:

The study population included 57 patients diagnosed with CKD caused by ON. They were referred to the Department of Nuclear Medicine for renal scan and radioisotope clearance to evaluate total and split renal function. All included patients were diagnosed to have CKD by means of clinical examination, biochemical analysis, and imaging abnormalities according to the K/DOQI criteria [12]. According to the documentation supplied with the referrals, the etiology of ON was as follows: urolithiasis (41% of the patients); intrinsic ureteric stricture (29% of the patients); ureteropelvic junction obstruction (20% of the patients); and unknown etiology (10% of the patients).

The control group consisted of 25 healthy individuals, potential kidney donors. Their renal function was evaluated by renal scan and clearance in accordance with the set algorithm for examination of renal function. They were referred to Department of Nuclear Medicine by Department of Nephrology and Clinical Immunology of Clinical Center of Vojvodina.

Point 2: The role of uromodulin has already been examined and correlated with histopathological data. Perhaps bibliographic entries should be mentioned (e.g. Urine Uromodulin as a Biomarker of Kidney Tubulointerstitial Fibrosis)

Response 2:

The mentioned reference (Melchinger, H.; Calderon-Gutierrez, F.; Obeid, W.; Xu, L.; Shaw, M.M.; Luciano, R.L.; Kuperman, M.; Moeckel, G.W.; Kashgarian, M.; Wilson, F.P.; et al. Urine Uromodulin as a Biomarker of Kidney Tubulointerstitial Fibrosis. Clin J Am Soc Nephrol 2022, 17, 1284-1292, doi:10.2215/cjn.04360422.) which focused on urinary Uromodulin was added to revised version of the manuscript – discussion section lines 326-329 (ref. no 33). We believed that pathohistological findings would add on the quality of our study which focused on serum Uromodulin.

Point 3: Moderate English changes required.

Response 3: English language and style were edited, and text was spell checked, as suggested.

Round 2

Reviewer 1 Report

I appreciate the authors have addressed my comments. I have no more comment.